# The Role of Selenoproteins SELENOM and SELENOT in the Regulation of Apoptosis, ER Stress, and Calcium Homeostasis in the A-172 Human Glioblastoma Cell Line

**DOI:** 10.3390/biology11060811

**Published:** 2022-05-25

**Authors:** Elena G. Varlamova, Michael V. Goltyaev, Egor A. Turovsky

**Affiliations:** Institute of Cell Biophysics of the Russian Academy of Sciences, Federal Research Center “Pushchino Scientific Center for Biological Research of the Russian Academy of Sciences”, 142290 Pushchino, Russia; goltayev@mail.ru

**Keywords:** selenium, selenoproteins, SELENOM, SELENOT, ER stress, calcium homeostasis, ERAD system, sodium selenite, methylseleninic acid, selenium nanoparticles

## Abstract

**Simple Summary:**

In this work, we present for the first time the effects of the suppression of the activity of poorly studied selenoproteins SELENOM and SELENOT in human glioblastoma cells, which is extremely important for understanding the functions of these proteins in brain cells. It has been shown that despite the structural similarity of these proteins, they affect the viability of these cancer cells in different ways, affecting various molecular mechanisms of regulation of pro-apoptotic genes, ER stress markers, and their physiological partners, as well as the regulation of cytosolic calcium.

**Abstract:**

It is known that seven mammalian selenoproteins are localized in the endoplasmic reticulum: SELENOM, SELENOT, SELENOF, SELENOK, SELENOS, SELENON, and DIO2. Among them, SELENOM and SELENOT are the least studied; therefore, the study of their function using the widespread method of suppressing the expression of genes encoding these proteins and the activity of the enzymes themselves by RNA interference is of great interest. We have shown that a decrease in the expression of SELENOM and SELENOT mRNA in the A-172 human glioblastoma cell line by more than 10 times and the quantitative content of enzymes by more than 3 times leads to ER stress, expressed as a decrease in the ER capacity for storing Ca^2+^ ions. At the level of regulation of apoptotic processes, SELENOM knockdown leads to an increase in the expression of pro-apoptotic CHOP, GADD34, PUMA, and BIM genes, but a compensatory increase in the levels of SELENOT and antioxidant genes from the group of glutathione peroxidases and thioredoxins did not induce cell death. Knockdown of SELENOT had the opposite effect, reducing the expression of pro-apoptotic proteins and regulating the level of a smaller number of genes encoding antioxidant enzymes, which also did not affect the baseline level of apoptosis in the studied cells. At the same time, ER stress induced by MSA or SeNPs induced a more pronounced pro-apoptotic effect in SELENOT knockdown cells through suppression of the expression of selenium-containing antioxidant proteins. Thus, in this work, for the first time, the mechanisms of fine regulation of the processes of apoptosis, cell proliferation, and ER stress by two ER resident proteins, SELENOM and SELENOT, are touched upon, which is not only fundamental but also applied to clinical importance due to the close relationship between the calcium signaling system of cells, folding proteins-regulators of apoptosis and cell survival pathways.

## 1. Introduction

Despite the fact that selenium (Se)-containing proteins were identified several decades ago, a third of them still remain poorly understood. The human selenoproteome is represented by 25 selenoproteins; the functions of half of them have not been established. It is known that seven of them are localized in the endoplasmic reticulum (ER): SELENOM, SELENOT, SELENOF, SELENOK, SELENOS, SELENON, and DIO2. The localization of these selenoproteins is closely related to the functions of this cell organoid; in particular, there is enough information indicating their participation in the regulation of ER stress caused by sources of various nature, including selenium-containing agents.

Among them, the least studied are the two selenoproteins SELENOM and SELENOT, belonging to the family of transmembrane proteins with thioredoxin-like folding; the main difference of which from the classical thioredoxin is the arrangement of secondary structure elements and the presence of additional α-helices, as well as the presence of selenocysteine in the active center of these proteins [1,2,3,4].

SELENOM has been identified using bioinformatic approaches and is a highly conserved protein across different animal species and classes. Human SELENOM (16.5 kDa) is expressed in many tissues but to a greater extent in the brain [5] and is most sensitive to Se deficiency in this organ. Thus, SELENOM can serve as a molecular biomarker of Se status and is involved in the regulation of human neurogenerative diseases [6]. Similar results in HT22 hippocampal and C8-D1A cerebellar cells were shown: stable overexpression of SELENOM prevents hydrogen peroxide-induced oxidative damage to cells [7]. Conversely, knockdown of the SELENOM gene in rats resulted in an increase in glutathione peroxidase and thioredoxin reductase activities in the brain, liver, lungs, and, to a lesser extent, in the kidneys and heart, while an increase in superoxide dismutase activity was observed only in the brain. Thus, SELENOM in different tissues can regulate the activity of two antioxidant enzymes in different ways. In addition, SELENOM overexpression in HT22 and C8-D1A cells increases the cytosolic calcium concentration in response to oxidative stress and may be involved in the regulation of apoptosis by blocking or delaying it [8].

SELENOT, like SELENOM, is a highly conserved protein and is localized exclusively in the ER. Human SELENOT (22.3 kDa) is 97–100% homologous to sequences of SELENOT in other species. This selenoprotein has a signal peptide (19 a.a.), as well as a highly hydrophobic region 16 a.a. at position 87–102 a.a., which is a transmembrane domain [9,10,11,12]. In neuroendocrine cells, SELENOT expression is regulated by the levels of cAMP and Ca^2+^ ions through the PACAP-peptide, which activates pituitary adenylate cyclase [10]. In addition, SELENOT expression is increased in neurons and astrocytes upon treatment with neurotoxin and oxidative stress, but the molecular mechanisms of regulation of such expression remain unknown. SELENOT has been found to have a neuroprotective effect during brain development and is necessary for proper behavior in adults, and it is involved in neurodegenerative diseases, which was demonstrated in mouse models of Parkinson’s disease and cell culture of dopaminergic neurons [13]. In addition, the important role of SELENOT in brain ontogenesis has been repeatedly shown: for example, a knockout line of mice in which SELENOT was specifically destroyed in nerve cells was characterized by a decrease in the volumes of various brain structures [14,15]. It has been shown that SELENOT interacts with keratinocyte-associated protein 2 (KSR 2) and other subunits of the OST complex, which may indicate its involvement in the regulation of N-glycosylation [16,17]. In addition, its relationship with GP78 (AMFR) and RMA1 (RNF5) proteins, which are actively involved in polyubiquitinylation and proteasomal degradation of proteins, has been shown [17].

Thus, according to the available data, it becomes clear that both selenoproteins are important for the normal development of the brain, and their expression in various brain cancer cells is quite high. For this reason, the main goal of this work was a comprehensive study of the consequences of SELENOM and SELENOT knockdown in A-172 human glioblastoma cells, including under conditions of ER stress caused by the action of Se-containing agents: methylseleninic acid (MSA), sodium selenite (SS), and selenium nanoparticles (SeNP). Previously, we have shown that the level of SELENOM and SELENOT mRNA expression and the amount of proteins changed depending on the concentration and nature of the ER stress source in various human cancer cell lines [18,19,20]. This effect was especially pronounced when cancer cells were treated with 5 µg/mL SeNP [21,22,23].

## 2. Materials and Methods

### 2.1. Isolation of RNA

Isolation of RNA from cell cultures was performed using a reagent for the isolation of total RNA-Extract tRNA reagent (Evrogen, Moscow, Russia) containing a solution of phenol and guanidine isothiocyanate. The reagent was added to a Petri dish with a cell monolayer at the rate of 1 mL per 10 cm^2^ of the surface, after which total RNA was isolated according to the manufacturer’s protocol. The quality of isolation RNA was checked by electrophoresis in 1% agarose gel, as well as on a spectrophotometer at a wavelength of 260 nm. To prevent contamination of genomic DNA, RNA samples were treated with DNase I at 37 °C for 1 h, after which the enzyme was inactivated by adding 50 mM EDTA to the mixture and heating to 60 °C for 10 min.

### 2.2. Reverse Transcription and Real-Time PCR

The reverse transcription reaction was carried out according to the protocol and using a kit for the synthesis of the first-strand cDNA (Evrogen, Moscow, Russia) containing reverse transcriptase MMLV. The reaction was carried out in the presence of oligo(dT) primers. The content of total RNA (1–2 µg) used in reverse transcription reactions was controlled by parallel amplification using primers specific to the reference gene. The resulting cDNA was used as a template for real-time PCR using the qPCRmix–HS SYBR mixture (Evrogen, Moscow, Russia). Amplification took place in the following temperature regime: 95 °C–1 min; 95 °C–10 s, 60 °C–10 s, 72 °C–15 s (35–40 cycles). In this case, the removal of fluorescence occurred during the elongation stage, the melting curve from 60 to 95 °C with a step of 0.3 °C. The results were expressed as 2^−∆(∆Ct)^, where Δ(ΔCt) is the difference between the ΔCt values between the studied gene and the reference gene. For real-time PCR, three human reference genes encoding β-actin, glyceraldehyde-3-phosphate dehydrogenase, and hypoxanthine phosphoribosyl transferase 1 were used. The sequences of all primers used in real-time PCR are shown in Table 1. To avoid the contribution of genomic DNA to the reaction result during cDNA amplification, we performed the selection of primers complementary to DNA regions located in different exons. The length of the amplified cDNA fragments was in the range of 150–300 bp. The quality of primer correct annealing was checked by DNA electrophoresis in 2% agarose gel, which was carried out after each RT-PCR run.

### 2.3. Western Blotting

Preparation of cell lysates for Western blotting was carried out by resuspending the study cells with a buffer containing 100 mM Tris-HCl, pH 8.0, 0.15 mM NaCl, 1 mM EDTA, 1 mM PMSF, at 4 °C, after which the lysates were centrifuged, and the supernatant was concentrated with using centrifuge concentrators Amicon Ultra 4–50 kDa (Merk Millipore) and analyzed by PAAG–electrophoresis in 10–12% resolving gel. Next, the proteins were electrotransferred from the gel to nitrocellulose or PVDF membranes. To prevent nonspecific binding of antibodies to the membrane, it was blocked with 5% defatted milk for 2 h at room temperature on a shaker, after which the membrane was washed three times with 1× PBST buffer (pH = 7.4) for 15 min at room temperature. Next, an immunodetection procedure was performed using primary antibodies against the label of interest or protein and secondary antibodies labeled with horseradish peroxidase. The dilution of primary antibodies was in the range of 1:100–1:300, and secondary antibodies were 1:5000–1:10,000; the membranes were incubated with each of the antibodies for 2–3 h at room temperature. Immunoreactive bands were visualized by determining peroxidase activity with DAB staining (0.05% DAB in TBS+10 µL 30% hydrogen peroxide). The following primary and secondary commercial antibodies purchased from Abcam and Invitrogene (USA) were used in the work: anti-GAPDH (#437000, Invitrogen, Waltham, MA, USA), anti-SELENOM (#PA5-72639, Invitrogen), anti-SELENOT (#PA5-26314, Invitrogen), anti-SELENOK (#PA5-34420), Invitrogen), anti-XBP1 (#PA5-27650, Invitrogen), anti-ATF-4 (#PA5-19521, Invitrogen), anti-ATF-6 (#PA5-20216, Invitrogen), anti-AMFR (#PA5-29501, Invitrogen), anti-RNF5 (#SAB4200208, Sigma-Aldrich, St. Louis, MO, USA), and secondary, conjugated with horseradish peroxidase (#656120, Invitrogen).

### 2.4. Gene Knockdown by RNA Interference, Transduction with Lentiviral Particles

Gene knockdown by RNA interference was performed using a vector containing a small hairpin RNA (shRNA) sequence, which was independently designed using the online resource https://www.invivogen.com/sirnawizard (accessed on 15 July 2021). For each of the genes, 3–5 shRNA variants and one negative control (scramble shRNA) were randomly selected to assess the specificity of knockdown and the side changes it causes in the cell. Delivery of shRNA within the pGPV vector containing the P1 promoter for shRNA expression was carried out by chemical transfection (lipofection) using the Lipofectamine 2000 reagent (Invitrogen, USA). Transfection efficiency analysis was performed using fluorescence microscopy since the pGPV plasmid contained the gene for the fluorescent CopGFP protein. To ensure the constant expression of shRNA in target cells, we used lentiviral particles. The construction of lentiviral vectors and the production of lentiviral particles were carried out at Evrogen, Moscow. Treatment of cancer cells with a preparation of lentiviral particles was carried out on 24-well plates at the stage of logarithmic cell growth with a confluence of 60–70%. In order to obtain a 100% transduced cell line, a 5-fold excess of lentivector relative to the number of cells was added (500 µL of lentivector with a titer of 106 IU/mL per 105 cells). After 24 h from the moment of transduction, the nutrient medium was changed. Since the pGPV lentiviral vector contained the gene for the fluorescent CopGFP protein, the result of transduction efficiency was assessed by fluorescent microscopy 72 h after transduction. Cells with lentiviral particles were further grown on a complete nutrient medium by selection with the antibiotic puromycin, the resistance gene, which is also localized in the pGPV vector. Evaluation of the efficiency of knockdown after transduction with lentiviral particles was carried out using real-time PCR and Western blotting. In our study, the following sequences proved to be the most effective: shRNA SELENOM (gctttcgtcacgcaggacattttcaagagaaatgtcctgcgtgacgaaagctttttt), shRNA SELENOT (gcgtgtgattaccagagaactattcaagagatagttctctggtaatcacacgtttttt).

### 2.5. MTT Analysis

MTT analysis was used by the MTT Cell Proliferation Assay Kit (Abcam, Cambridge, UK). To do this, cells were grown on 96-well plates up to approximately 5500 cells per well and treated with the study agent. After which 20 μL of MTT working solution (5 mg/mL) in sodium phosphate buffer (pH 7.4) was added, carefully pipetted, and incubated 3 h at 37 °C and 5% CO_2_. Next, the medium was removed and the cells were slightly dried. For fast and uniform dissolution of formazan, 200 µL of DMSO was added to each well and incubated on a plate shaker for 10 min at room temperature. Next, using a plate reader, the optical density of solutions was measured in each test well at OD = 570 nm and background values at OD = 670 nm. This procedure was repeated three times.

### 2.6. Ability of Cells to Proliferate on Soft Agar

A stock solution of agarose (3.2%) was autoclaved and cooled in a water bath to 38.5 °C. Next, it was used to first prepare the base layer of agarose (0.8%): 1.75 mL of agarose stock solution was mixed with 5.25 mL of culture medium per 30 mm dish. After careful mixing with a pipette, 1 mL of agarose was poured into dishes and cooled for 5–10 min at 4 °C until it solidified. Next, the top layer of agarose (0.48%) was prepared: the cells, previously removed from the dishes with trypsin, were resuspended in 5 mL of the culture medium, after which they were mixed with 750 µL of agarose stock solution (38.5 °C), carefully pipetted, and 1 mL per dish was added over the base layer. After the top layer of agarose solidified at 4 °C for 5–10 min, 1 mL of culture medium was added to the dish on top of it. The cells were incubated in a CO_2_ incubator for 9–12 days until colony formation, changing the medium every three to four days. Staining of the grown colonies was carried out as follows: 1 mL of PBS containing 4% formaldehyde and 0.005% Crystal violet were added over the top layer of agarose, draining the culture medium beforehand, incubated for 2 h, rinsed the dishes from the dye, and counted the colonies.

### 2.7. Wound Healing Assay

For the “wound healing assay”, disruption of the structure of the cell monolayer was performed by pushing the monolayer with a pipette tip, which was like a “wound”. Control of cell migration to the “wound” area using a microscope was carried out after 24, 48, and 72 h.

### 2.8. Registration of Changes in Cytosolic Ca^2+^

To detect changes in [Ca^2+^]_i_, cancer cells were grown at round coverslips in DMEM medium supplemented with 10% fetal calf serum for 48 h in a CO_2_ incubator to confluency 80–90%. Experiments on recording the level of cytosolic Ca^2+^ in cells loaded with a Fura-2AM fluorescent calcium probe. A-172 cells were loaded with the probe dissolved in Hanks balanced salt solution (HBSS) composed of (mM): 156 NaCl, 3 KCl, 2 MgSO_4_, 1.25 KH_2_PO_4_, 2 CaCl_2_, 10 glucose, and 10 HEPES, pH 7.4 at 37 °C for 40 min with subsequent 15 min washout. The final concentration of Fura-2AM was 5 µM. Experiments were carried out in Hanks’ solution and 10 mM HEPES, pH 7.4. To register changes of [Ca^2+^]_i_, we used an inverted motorized fluorescent microscope, Axiovert 200M (Carl Zeiss Microscopy GmbH, Jena, Germany), with a high-speed monochrome CCD-camera AxioCam HSm (Carl Zeiss Microscopy GmbH, Jena, Germany), and a high-speed light filter replacing system Ludl MAC 5000 (Ludl Electronic Products, Hawthorne, NY, USA). The reagents were added and washed using the perfusion system that provides a perfusion rate of 15 mL/min. For Fura-2 excitation and registration, we used the objective HCX PL APO 20.0 × 0.70 IMM UV, refraction index 1.52. Camera settings are 500 pixels × 500 pixels (Voxel Size 0.724 µm × 0.723 µm), binning 2 × 2, resolution 14 bits. Emission of Fura-2 stained cells was recorded upon excitation at the wavelengths of 340 and 387 nm. The frame rate was 1 frame per 3 s. The resulting two-channel (when Fura-2 was excited at 340 and 387 nm) time-lapse series of images were processed with ImageJ software (RRID: SCR_003070, NIH Image, Bethesda, MD, USA). Background fluorescence was subtracted frame by frame using the Math Subtract plugin in ImageJ. All experiments were carried out at 28–30 °C.

### 2.9. Statistical Analysis

Microsoft Excel and GraphPadPrism 5 software were used to analyze the data, create graphs, and process statistics. Protein evaluation was carried out in different samples according to the Lowry method. The protein concentration was calculated from a standard curve constructed using 1 mg/mL BSA solution. Values were given in the work as the mean ± standard deviation of at least three independent experiments. Differences were considered significant at *p* < 0.05. Protein expression was quantified using ImageJ software. Origin 8.5 (Microcal Software Inc., Northampton, MA, USA) and Prism 5 (GraphPad Software, La Jolla, CA, USA) were used for plotting and statistical processing. The significance of differences between groups of experiments was determined using Student’s *t*-test and within groups—Student’s *t*-test. Differences were considered significant: *** at *p* < 0.001, ** at *p* < 0.01, * at *p* < 0.05, n/s—differences were not significant.

## 3. Results

### 3.1. SELENOM-KD and SELENOT-KD Do Not Significantly Affect the Proliferative Properties of A-172 Cells

Since, according to previously obtained data [5,6,7,8,10], mRNA expression of the SELENOM and SELENOT genes is quite high in the brain, we chose human glioblastoma cells (line A-172) to achieve the most pronounced knockdown of these selenoproteins. First of all, three vectors were constructed for each of the two genes carrying different shRNA sequences. Transfection of A-172 cancer cells with each of the constructs made it possible to identify the most effective of them. Next, lentiviral particles containing the SELENOM-KD and SELENOT-KD were prepared. We obtained two variants of the A-172 cell line stably transduced with the studied knockdown constructs by selection with puromycin (0.5 µg/mL). As a result of this series of experiments, two variants of human glioblastoma cancer cells were obtained. A-172 cells were loaded with Ca^2+^-sensitive probe Fura-2AM to determine the density of cell culture, and to determine the effectiveness of SELENOM-KD and SELENOT-KD, cells were imaged in the eGFP fluorescence recording channel. According to the presented photographs, one can speak of the high efficiency of cancer cell transduction (Figure 1a,b). PCR data showed that the level of mRNA expression of the genes encoding SELENOM and SELENOT decreased by 10 times (Figure 1c), while the level of each of the selenoproteins decreased by approximately 3 times compared to the control (scraRNA) (Figure 1d,e; Appendix A).

Since the lentiviral constructs we created significantly suppressed the expression of SELENOM and SELENOT, this allowed us to study the mechanisms of action of these proteins in cancer cells. According to the results obtained using MTT analysis, we can speak about the absence of significant differences in the proliferative properties of control cells and cells with SELENOM-KD or SELENOT-KD (Figure 1f).

### 3.2. SELENOM-KD and SELENOT-KD Do Not Contribute to the Acquisition of Signs of Normal Cells by A-172 Cancer Cells

To date, the important role of various selenoproteins in carcinogenesis is obvious, and experiments are often carried out by reducing or overexpressing their activity to understand the role of these proteins in the oncotransformation of normal cells or the reverse process for cancer cells [18,19,20]. In this regard, we carried out a series of experiments aimed at elucidating the role of SELENOM-KD and SELENOT-KD in the acquisition by A-172 cancer cells of signs of normal healthy cells. It is known that the degree of collective cell migration is eliminated in the cell culture of many types of cancer. It is assumed that collective cell migration is an in vitro similarity of the mechanism that operates in vivo to ensure normal tissue homeostasis, and it is suppressed during tumor formation. To analyze this ability, we disrupted the structure of the cell monolayer by dragging it with a pipette tip, which was similar to a wound. Cell migration to the “wound” area was monitored using a microscope after 24 and 48 h. It can be seen that there were no significant differences in the control sample and separately in the samples with SELENOM-KD (Figure 2a,b) and SELENOT-KD (Figure 2c,d).

Next, we tested the ability of A-172 cells with and without KD to proliferate on soft agar since it is known that the formation of colonies on soft agar is one of the strict tests of their malignant transformation (Figure 3a–c). In order to test for the presence of an inverse effect of SELENOM-KD (Figure 3a) and SELENOT-KD (Figure 3b) on the ability of glioblastoma A-172 cells to proliferate on soft agar, the cells were plated on soft agar medium and incubated for several weeks, periodically changing the growth medium. According to the results of this test, shown in Figure 3, we can speak about the absence of this knockdown effect of each of the selenoproteins on the growth of cancer cells without the additional influence of external stimuli on soft agar.

Thus, SELENOM-KD and SELENOT-KD, in the absence of external stimuli on A-172 human glioblastoma cells, did not affect the proliferative activity of cells, their transformation into normal cells, migration, and the formation of cell colonies.

### 3.3. SELENOM-KD and SELENOT-KD Lead to Impaired Activity and Ca^2+^ Capacity in the ER

It is known that both selenoproteins are involved in the regulation of calcium homeostasis in various cells [8,10], so it was important to follow how this regulation changes with a decrease in the activity of these proteins in human glioblastoma cells. To do this, A-172 cells were loaded with Ca^2+^-sensitive probe Fura-2AM (Figure 1a,b). Application of 10 μM thapsigargin (TG), an inhibitor of the sarco/endoplasmic reticulum Ca^2+^ ATPase (SERCA), in a calcium-free medium with the addition of 0.5 mM Ca^2+^-chelator EGTA, caused an increase in the concentration of cytosolic Ca^2+^ ([Ca^2+^]_i_) in A-172 cells. In response to the application of thapsigargin, there was a rapid increase in [Ca^2+^]_i,_ followed by a gradual pumping of Ca^2+^ ions from the cytosol to a new steady state as in control cells (Figure 4, Scra) and in cells with SELENOM-KD (Figure 4a). In the case of SELENOT-KD, the [Ca^2+^]_i_ removal process had slow kinetics and did not occur until the resting level (Figure 4b), which may indicate the involvement of SELENOT in the regulation of PMCA activity.

However, the most significant effect of SELENOM-KD and SELENOT-KD was directed to the Ca^2+^ capacity of the ER. Since TG was added in a calcium-free medium and there was no influx of Ca^2+^ ions from outside, the amplitude of the increase in [Ca^2+^]_i_ level reflected only the Ca^2+^ capacity of the ER. The amplitudes of the Ca^2+^ signals to the addition of TG in cells with SELENOM-KD (Figure 4a) and SELENOT-KD (Figure 4b) were almost identical at the maximum of the Fura-2 fluorescence signal but 76% lower than in the control (Figure 4, Scra).

Thus, SELENOM-KD and SELENOT-KD contributed to functional disorders of the ER, reducing its capacity for storing Ca^2+^ ions, with the most pronounced effect being shown for SELENOM-KD when the ER was practically empty and the application of TG led to a mild increase in [Ca^2+^]_i_. Such an effect on the endoplasmic reticulum can significantly alter the expression of proteins that regulate cell viability pathways.

### 3.4. SELENOM-KD and SELENOT-KD Affect Expression Patterns of Pro-Apoptotic Genes, ER Stress Markers, and Selenoproteins in Different Ways

When studying the effect of SELENOM-KD on the expression of pro-apoptotic genes, we found an increase in mRNA expression of a number of pro-apoptotic genes: CHOP (3.5 times), GADD34 (2.7 times), PUMA (2.6 times), BIM (3 times). At the same time, mRNA expression of other genes involved in the activation of various signaling pathways of apoptotic cell death was not detected (Figure 5a).

When establishing the association of SELENOM-KD with the activation of three known UPR pathways (PERK, IRE1α, and ATF6) in the studied cancer cell line, we showed no effect of SELENOM-KD on the expression patterns of three key markers of these signaling pathways. According to the real-time PCR results shown in Figure 5b, expression levels of the ATF-4, XBP1s, and ATF-6 did not change significantly during SELENOM-KD. Since SELENOM and SELENOT are residents of the ER, it was important to follow how the mRNA expression of other ER selenoproteins would change under the same conditions. According to the results presented in Figure 5c, in A-172 cells with SELENOM-KD, the expression of DIO2 was significantly reduced by more than two times, and there was also an increase in the expression of SELENOT and SELENOK by 2.5 and 2.2 times, respectively.

In addition, it is known that Se-containing enzymes glutathione peroxidase and thioredoxin reductase are the key regulators of the redox status in the cell. Therefore, we studied changes in the expression of mRNA levels of three thioredoxin reductases and four glutathione peroxidases under SELENOM-KD. According to the results shown in Figure 5d, some of these enzymes were upregulated in SELENOM-KD cells compared to controls. Thus, a more than 2-fold increase in mRNA expression was observed for GPX1 and GPX2, as well as for TXNRD3.

When studying the consequences of SELENOT-KD in A-172 cells, we performed a similar series of real-time PCR, which resulted in the opposite effect of SELENOT-KD on the expression of a number of pro-apoptotic genes. Thus, there was a decrease in expression levels for the genes CHOP, GADD34, PUMA, and BIM to the following values 0.3, 0.6, 0.2, and 0.6, respectively (Figure 6a).

Similar to SELENOM-KD, downregulation of SELENOT mRNA in A-172 cells did not significantly affect the expression of ER stress markers (Figure 6b).

Interestingly, SELENOT-KD significantly affected the expression of the same ER resident selenoproteins (Figure 6c). According to the obtained results, the expression level of SELENOK and SELENOM mRNA increased by 2 and 3.7 times, respectively, which is consistent with the results of Western blotting (Figure 7a,b; Appendix A). However, unlike SELENOM-KD, SELENOT-KD practically did not affect the mRNA expression of the studied glutathione peroxidases and thioredoxin reductases (Figure 6d).

Thus, SELENOM-KD in A-172 cells resulted in increased expression of a number of key pro-apoptotic genes and two selenoproteins, SELENOT and SELENOK. On the other hand, SELENOM-KD did not significantly affect the expression patterns of Se-containing glutathione peroxidases and thioredoxin reductases, as well as key markers of the three ER stress signaling pathways. In turn, SELENOT-KD affected the expression of the same pro-apoptotic genes in the opposite way, reducing their expression. At the same time, SELENOT-KD changed the expression of ER resident selenoproteins and did not affect the expression of mRNA of ER stress markers, such as SELENOM-KD. In addition, SELENOT-KD had almost no effect on the mRNA expression of the studied glutathione peroxidases and thioredoxin reductases.

### 3.5. SELENOM-KD and SELENOT-KD Do Not Significantly Affect the Proliferative Properties of A-172 Cells under ER Stress

Previously, we and other authors have shown that methylselenic acid (MSA), sodium selenite (SS), and selenium nanoparticles (SeNP) can affect the expression of SELENOM and SELENOT mRNA in various cancer and normal cells, depending on the concentration and time of exposure [19,20,21,22,23,24,25,26,27,28]. It was shown that SS at a concentration of 0.1 µM did not significantly affect the expression of mRNA of both selenoproteins, just as 0.1 µM MSA did not affect the expression of SELENOM mRNA. In contrast, significant increases in expression were recorded for the SELENOT gene when cells were treated with 0.1 μM MSA. In addition, SeNP at a concentration of 5 μg/mL contributed to a sharp increase in the expression of mRNA of both selenoproteins. All three ER stress inducers at the indicated concentrations and after 24 h of exposure led to acute ER stress and increased apoptotic death of A-172 cells [19,20,22]. Therefore, to determine the effect of these selenium agents on the activity of mitochondrial reductases in living cells, which makes it possible to indirectly judge the change in their proliferative properties in the presence or absence of SELENOM-KD or SELENOT-KD, we performed an MTT analysis.

According to the results of the MTT analysis presented in Figure 8a,b SELENOM and SELENOT under ER stress caused by 24 h exposure of A-172 cells to 0.1 μM concentrations of MSA or SS, as well as 5 μg/mL SeNP, did not significantly affect the change in the proliferative properties of these human cancer cells.

### 3.6. SELENOM-KD and SELENOT-KD Induce Changes in the Expression Patterns of Pro-Apoptotic Genes, ER Stress Markers, and Selenoproteins under Conditions of ER Stress Caused by Various Selenium-Containing Inducers

In addition, it was important to understand how SELENOM-KD and SELENOT-KD can affect the expression of the studied genes under acute ER stress. In studying the effects of SELENOM-KD on expression patterns of pro-apoptotic genes, key ER stress markers, and selenoproteins under conditions of acute ER stress, one ER stress inducer, SeNP, was selected at a concentration of 5 µg/mL. According to the results of real-time PCR, under conditions of 24 h exposure to 5 µg/mL SeNP on cells and simultaneously SELENOM-KD, an increase in expression levels of mRNAs of the pro-apoptotic genes encoding CAS-3 (by 2.6 times), CAS-4 (57%), MAP3K5 (by 4.9 times), and MAPK-8 (by 2.3 times) was observed (Figure 9a).

In addition, increased expression was also characteristic of some markers of ER stress: ATF-4 and XBP1s. According to the results of RT-PCR, as well as Western blotting (Figure 9b and Figure 10a,b; Appendix A), we can speak of an increase in ATF-4 mRNA by more than three times and XBP1s by more than two times. This increase in expression was duplicated by the results of Western blotting. At the same time, the expression of the marker of the third signaling pathway UPR-ATF-6 practically did not change.

In addition, under these conditions, SELENOM-KD did not significantly affect the expression of other ER resident selenoproteins (Figure 9c) and thioredoxin reductases, while the expression of glutathione peroxidase 3 decreased by 1.5–2 folds (Figure 9d).

When studying the effect of SELENOT-KD on the expression patterns of the above genes, it was also important to take into account the conditions of acute ER stress caused by 0.1 μM MSA since this inducer increased mRNA expression in A-172 cells by more than two times [19]. According to the results of real-time PCR, SELENOT-KD after 24 h of cell exposure to 0.1 μM MSA also contributed to a more than 2-fold decrease in the expression of a number of pro-apoptotic genes (CHOP, PUMA, BIM, GADD34). However, no significant changes in mRNA expression of other studied genes were observed (Figure 11a–d).

After 24 h of treatment of cells with SeNP at a concentration of 5 μg/mL, a decrease in pro-apoptotic genes by more than 2–3 times was also observed; this was typical for the genes encoding CHOP, GADD34, PUMA, BIM, occurring against the background of an increase in the MAP3K5 (by 3.2 times) and MAPK-8 (by 4.9 times) genes (Figure 11a). Interestingly, ER stress in A-172 cells with SELENOT-KD altered ATF-4 and ATF-6 expression, and SeNP altered XBP1s expression (Figure 11b), which may be related to different signaling pathways activated by ER stress inducers. At the same time, almost no changes were observed in the mRNA expression of genes encoding other selenoproteins and enzymes that regulate the oxidative status of cells (Figure 11c,d).

### 3.7. SELENOT-KD Causes a Decrease in the Expression of AMFR and RNF5 mRNA in A-172 Cells Both under Conditions of ER Stress and without It

To date, sufficient information has been accumulated regarding the physiological partners of SELENOT in various cancerous and normal cells of the brain and other tissues and organs. Based on these data, it becomes clear that SELENOT is able to interact with various subunits of the OST complex, in particular, with KCP2, STT3A, OST48, and also with ubiquitin protein ligases GP78 and RMA1 [14,15]. The proteins of the OST complex are known to be involved in the retrotranslocation of certain misfolded proteins. In addition, it is known that there are Derlins (Derlins 1–3), the main function of which perform a similar function and are involved in the regulation of ER stress. Therefore, in this work, we also investigated whether SELENOT-KD affects the expression of this family of proteins (Figure 12b,c; Appendix A). According to the real-time PCR results shown in Figure 12a, it can be concluded that SELENOT-KD has a significant effect on the mRNA expression of ubiquitin protein ligases AMFR (GP78) and RNF5 (RMA1). Thus, the expression of AMFR (GP78) and RNF5 (RMA1) in A-172 cells, both under conditions of ER stress caused by exposure to cells of 0.1 μM MSA and SeNP at a concentration of 5 μg/mL, and without it, decreased by more than two times. Similar results were shown by Tanguy Y. et al. on endocrine cells [16].

## 4. Discussion

The focus of this work was the study of the consequences of SELENOM-KD and SELENOT-KD, two ER resident selenoproteins, in human glioblastoma cells (line A-172). The novelty of this study is the identification of the shRNA sequence, the use of which led to a decrease in the mRNA expression of the SELENOM gene of this protein by almost 10 times and the quantitative content of the protein by almost 3 times. Perhaps such a pronounced effect of suppressing the expression of the studied selenoproteins is explained by the choice of the A-172 cell line since it is known that the mRNA expression of both genes is quite high in this cell line and other brain cells [5,6,7,8,10].

In the work, the opposite effect of the influence of SELENOM-KD and SELENOT-KD on the levels of expression of pro-apoptotic genes: CHOP, GADD34, PUMA, and BIM was established. In the case of SELENOM-KD, an increase in mRNA expression of these genes by more than two times was observed, while with SELENOT-KD, on the contrary, a decrease was observed. It is well known that CHOP acts as a transcription factor that, in combination with ATF-4, enhances the expression of various genes. Under conditions of ER stress, active proteins containing only the BH3-domain, which include BIM and PUMA, trigger the oligomerization of pro-apoptotic BAK and BAX proteins, which ultimately leads to mitochondrial permeabilization and release of cytochrome C into the cytosol. After that, the apoptosomal complex is formed, and the activation of the caspase cascade is triggered, which ultimately leads to mitochondria-mediated apoptosis [29,30]. However, in our experiments, SELENOM-KD did not cause an increase in the gene. Similar results on increased expression of pro-apoptotic genes with reduced SELENOM activity were demonstrated in HT22 hippocampal cells and C8-D1A cerebellar cells [7]. With stable overexpression of this selenoprotein, the prevention of oxidative damage to cells caused by hydrogen peroxide was observed, while SELENOM-KD led to an increase in glutathione peroxidase and thioredoxin reductase activities in the brain, liver, lungs, and, to a lesser extent, in the kidneys and heart [7]. Thus, there is a number of strong evidence indicating the involvement of SELENOM in the activation of protective antioxidant mechanisms in brain cells, regardless of whether they are cancer or healthy cells. In addition, in our experiments, an increase in mRNA expression of two glutathione peroxidases (GPX1 and GPX2), which have a wide substrate specificity, including hydrogen peroxide, tert-butyl hydroperoxide, hydroperoxidesas [31]. Moreover, one of the three thioredoxin reductases is TXNRD3, which has not only thioredoxin reductase activity but also glutathione reductase and glutaredoxin activities in the presence of NADPH, were also observed [32]. Activation of these antioxidant enzymes may be one of the reasons for the lack of effect of SELENOM-KD on the proliferative properties of A-172 cancer cells with an increase in the expression of pro-apoptotic genes and the prevention of apoptosis, as evidenced by the absence of changes in the expression of mRNA of other pro-apoptotic genes (CAS-3, CAS-4, BAX, BAK, MAPK-8, MAP3K5), as well as markers of ER stress signaling pathways.

Another possible explanation for the lack of influence of SELENOM-KD on the proliferative properties of cancer cells is the increased expression of SELENOK and SELENOT, two other selenoproteins localized in the ER. According to previous data, both proteins are involved in the regulation of post-translational modification of proteins and degradation of misfolded proteins. So it is known that Derlin-1, Derlin-2, p97, SELENOS, as well as components of the OST complex: ribophorins I and II, OST 48, STT3A, ER chaperones, and calnexin were found as partners of SELENOK. Thus, SELENOK is involved in the regulation of the ERAD system and is involved in maintaining ER homeostasis [33,34]. Similar data were also obtained for SELENOT, which was discussed earlier in this work. In addition, it is known that SELENOK, along with SELENOS, is involved in the control of the transfer of vasoline-containing p97(VCP) protein to the ER membrane during ERAD [35]. This interaction with p97(VCP) is also necessary for the regulation of ER stress and degradation of ERAD substrates such as, for example, CD3δ [36]. In addition, it is known that SELENOK has peroxidase activity and is able to restore harmful hydrophobic substrates, such as phospholipid hydroperoxides, by participating in the restoration of the bilipid layer of the membrane [37].

Thus, increased expression of SELENOK and SELENOT mRNAs under conditions of suppressed SELENOM activity presumably contributed to the activation of the ERAD system and a decrease in the number of misfolded proteins. This, in turn, caused a mild adaptive response of the cell in the form of increased expression of a number of pro-apoptotic genes, which did not significantly affect redox homeostasis in A-172 cells under SELENOM-KD.

Under SELENOT-KD conditions, most likely, a similar situation could occur since an increase in the activity of the selenoproteins SELENOK and SELENOM was observed. Despite the fact that, in this situation, no increase in the expression of glutathione peroxidases and thioredoxin reductases was observed, it is possible that overexpression of SELENOM could cause a decrease in the expression of a number of pro-apoptotic genes listed above. Here it is rather difficult to explain the reason for the decrease in their expression: either this occurred due to a decrease in SELENOT activity or as a result of increased SELENOM activity. The second version is supported by the fact that there are a sufficient number of works indicating the opposite effect of SELENOT-KD on various mammalian cells. Thus, it was shown that in mouse pituitary cancer cells (AtT20 cell line), a decrease in SELENOT activity, on the contrary, contributed to an increase in oxidative and ER stress, which led to a decrease in the viability of these cells [15]. On dopaminergic neurons, in cortical neuroblasts, and in PC12 cells (cell line derived from rat adrenal medulla pheochromocytoma), SELENOT-KD promoted increased oxidative stress [38,39,40,41]. In addition, SELENOT has been shown to protect the heart from oxidative stress and ischemia after ischemia/reperfusion in rats [42]. An additional explanation for why SELENOT-KD did not lead to the death of A-172 cancer cells and did not cause either oxidative stress or ER stress can be the increased activity of SELENOK. Indeed, from the previously obtained data, the SELENOT and SELENOK functions overlap.

Thus, based on the obtained results, it is possible to suggest some functional relationship between the selenoproteins SELENOT, SELENOK, and SELENOM in the regulation of the levels of expression of pro-apoptotic genes, as well as in the operation of the ERAD system, however, other independent approaches are required to test this hypothesis.

The second large series of experiments in this work was devoted to the study of the role of SELENOM-KD and SELENOT-KD in the regulation of ER stress caused by three selenium-containing agents of different nature: 0.1 µM SS, 0.1 µM MSA, and 5 µg/mL SeNP. Previously, we have repeatedly shown that the expression of the studied selenoproteins can change differently when cancer cells are exposed to these ER stress inducers, depending on the concentration, exposure time, and cancer cell line [19,23]. In this work, it was shown that SELENOM-KD and SELENOT-KD under conditions of ER stress also did not significantly affect the proliferative properties of A-172 cancer cells, even when cells were exposed to 5 µg/mL SeNP for 24 h. Previously, we showed that 5 µg/mL SeNP caused an increase in the expression of mRNA of these selenoproteins in three of the studied cancer cell lines, including A-172, by more than three times [23], and therefore we expected the opposite effect.

In addition, upon exposure of A-172 cells to SeNP at a concentration of 5 μg/mL for 24 h, an increase in expression levels was observed for almost all pro-apoptotic genes, but the strongest effect was characteristic of the genes encoding CAS-3 (by 2.6 times), MAP3K5 (by 4.9 times), MAPK-8 (by 2.3 times). Activation of ATF-4 and XBP1s markers of PERK and IRE1α UPR pathways was also observed. A similar pattern of changes in expression patterns of pro-apoptotic genes and UPR markers was observed earlier by us after 24 h of exposure to A-172 cells with 5 µg/mL SeNP [23]. ER stress in these cancer cells leads to activation of MAP kinases, which can activate apoptotic signaling by upregulating pro-apoptotic genes through transactivation of specific transcription factors or by directly modulating the activity of mitochondrial pro- and anti-apoptotic proteins through various phosphorylation events. Since we have repeatedly shown that SS, MSA, and SeNP are able to induce apoptotic death of cancer cells of various lines, an increase in the expression of MAP kinases, in this case, enhances the activation of apoptosis signals, which is also evidenced by an increase in the expression of CAS-3 mRNA changes neither in the expression of ER resident selenoproteins nor in glutathione peroxidases and thioredoxin reductases.

In the study of the effect of SELENOT-KD on the expression patterns of the above genes under conditions of ER stress caused by 24 h exposure to cells of 5 µg/mL SeNP and 0.1 µM MSA, a decrease in the expression of mRNA of a number of pro-apoptotic genes (CHOP, PUMA, BIM, GADD34) by more than 2–3 times. However, no significant changes in mRNA expression of other studied genes were observed. Like SELENOM-KD, an increase in the expression of genes encoding MAP3K5 and MAPK-8 by more than 3–4 times was observed.

There is an opinion that the activation of MAP kinases is the result of the influence of Se-containing sources of ER stress on cells, and the increase in their expression is directly related to the consequences of the activity suppression of the proteins. However, neither SELENOM-KD nor SELENOT-KD led to significant changes in the proliferative properties of A-172 cells under conditions of ER stress.

An important part of the work was devoted to the study of changes in mRNA expression patterns and the activity of the proteins themselves, which were previously identified as possible functional and physical partners of SELENOT since there is practically no such information regarding SELENOM. In the work, it was found that SELENOT-KD in A-172 cells contributed to a decrease in the expression of mRNA of the genes encoding AMFR and RNF5 proteins, which was also confirmed using Western blotting. Ubiquitin protein ligase AMFR-E3 mediates polyubiquitinylation of lysine and cysteine residues of target proteins for subsequent proteasomal degradation. It was originally identified as an autocrine tumor motility factor receptor that promotes invasion and metastasis of tumors. RNF5 is an E3 ubiquitin protein ligase, and RNF5-KD significantly reduces AMFR-mediated ubiquitinylation of CFTR, a transmembrane conductance regulator in cystic fibrosis [43]. AMFR and RNF5 can be linked to each other through members of the Derlins family (Derlin 1, 2, 3) and function as a Derlin-containing complex for polyubiquitinylation of a number of proteins. It has been shown that SELENOT is involved in N-glycosylation of endogenous glycoproteins and is a subunit of the A-type oligosaccharyl transferase complex (OST) [11]. The results of this work are consistent with the previously revealed dependence of a decrease in AMFR and RNF5 mRNA expression on a decrease in SELENOT mRNA expression [11]. It is known that SELENOT can interact with KRTCAP2 (KCP2), STT3A, DDOST (OST 48), but not with STT3B [11]. However, according to our results, mRNA expression of these proteins did not practically change during SELENOT-KD both under ER stress conditions and in intact cells. It has been reported that the downregulation of two important enzymes of the ERAD system in SELENOT-KD results in the accumulation of misfolded proteins in the ER lumen, which appears to exacerbate ER stress in A-172 cancer cells.

It is known that the role of Ca^2+^ ions in the regulation of physiological and pathophysiological processes in the body is very high [44,45]. Ca^2+^ ions can enter the cell from the extracellular medium through channels of the plasma membrane and via mobilization of Ca^2+^ ions from the ER. It is known that SELENOM and SELENOT are ER resident selenoproteins and are involved in the regulation, including the calcium component of intracellular cell signaling [7,10]. There is evidence that overexpression of SELENOT in PC12 cells promotes an increase in Ca^2+^ signals in response to a stimulus, while knockdown of this gene, on the contrary, inhibits Ca^2+^ signals when cells are activated by the PACAP neuropeptide (pituitary adenylate cyclase-activating polypeptide) and reduced hormone secretion [10].

At the same time, it has been shown for SELENOM that its overexpression protects brain cells from H_2_O_2_-mediated intracellular Ca^2+^ flux and death, while SELENOM-KD leads to an increase in cytosolic Ca^2+^ levels, enhanced oxidative stress, and apoptosis [7]. The functions of SELENOM and its relationship with calcium signaling in cancer cells have not been studied, but based on our data, it can be assumed that SELENOM-KD leads to an increase in the Ca^2+^ capacity of the ER, which correlates with an increase in the expression of pro-apoptotic genes, promoting the initiation or enhancement of apoptosis.

Thus, despite the fact that both selenoproteins SELENOT and SELENOM belong to the same family of membrane proteins with thioredoxin-like folding and are localized in the ER, they have opposite effects on the expression of a number of pro-apoptotic genes in glioblastoma cells. It was found in the work that SELENOM-KD promotes an increase in the expression of a number of pro-apoptotic genes, and SELENOT-KD, on the contrary, a decrease, including under conditions of ER stress. At the same time, SELENOM-KD and SELENOT-KD similarly affected the expression of the same selenoproteins; in addition, the silencing of each of them led to an increase in the expression of the other. This may indicate a functional relationship between these selenoproteins in the ER, as well as the regulation of the activity of another ER resident, SELENOK. The work also found that neither SELENOM-KD nor SELENOT-KD significantly affected the proliferative properties of glioblastoma cells, including those under ER stress. This may be explained by the fact that the changing expression of a number of important pro-apoptotic genes, selenoproteins, MAP kinases, Se-containing thioredoxin reductases, and glutathione peroxidase contributed to the maintenance of redox homeostasis in knockdown cells in the absence of exposure to sources of ER stress. In addition, a decrease in the activity of both selenoproteins did not significantly affect the proliferative properties of cancer cells, which were significantly reduced by the action of Se-containing ER stress inducers studied in this work. Previously, it was repeatedly shown that SELENOT is involved in the regulation of the ERAD system through physical interaction with the components of the OST complex, as well as participation in the processes of N-glycosylation of proteins with disulfide bridges, for example, glycoprotein hormones with disulfide bonds [16]. In addition, it was shown that SELENOT is required for the attachment of glycosylphosphotidylinositol to the C-terminus of proteins during their post-translational modification, which is extremely important for the efficient processing of more than 150 proteins [17]. This may indicate that with a decrease in SELENOT activity, proteins with incorrect folding can accumulate in cells due to dysfunction of the ERAD system; however, increased expression of another selenoprotein, SELENOK, which is also shown to be involved in the regulation of the ERAD system, compensates for the decrease in SELENOT activity, as well as two other proteins AMFR and RNF5, which are polyubiquitin ligases [43]. It is possible that SELENOT directly or indirectly interacts with thiol groups and/or glycosylated regions of intracellular calcium channels and pumps and regulates their activity through a redox mechanism. This hypothesis is increasingly supported by studies on the involvement of other ER resident selenoproteins (SELENOK, SELENOM) in the regulation of calcium homeostasis and the functioning of calcium channels [18,19,20]. In addition, strong evidence in the regulation of calcium homeostasis of both selenoproteins in A-172 cells was also demonstrated in this work. Schematically, the activation of key enzymes, as well as processes caused by the suppression of the activity of two important ER resident selenoproteins studied in this work, is shown in Figure 13.

## 5. Conclusions

In this work, for the first time, we studied changes in the redox status, calcium homeostasis, and expression of a number of key pro-apoptotic genes and ER stress regulators under conditions of reduced activity of two selenoproteins SELENOM and SELENOT in human glioblastoma cells (A-172). The opposite effect of SELENOM and SELENOT knockdowns on the expression of pro-apoptotic genes was revealed. It can be assumed with a high probability that SELENOM has anti-apoptotic properties, while SELENOT, on the contrary, is pro-apoptotic; however, more experiments are needed to investigate cell viability and rate of apoptosis over time. The relationship of knockdown in the regulation of the expression of its two functional partners (AMFR and RNF5), which play an important role in the proteasome degradation of proteins with misfolding, has been established. It was shown that SELENOM-KD under conditions of ER stress caused by 5 µg/mL SeNP led to the activation of PERK and IRE1α UPR pathways, as well as MAP kinases. It was found that the suppression of the activity of each of the selenoproteins did not significantly affect the proliferative properties of cancer cells and also did not lead to the appearance of signs of normal cells in these cancer cells. SELENOM-KD and SELENOT-KD lead to impaired activity and Ca^2+^ capacity in the ER.

## Figures and Tables

**Figure 1 biology-11-00811-f001:**
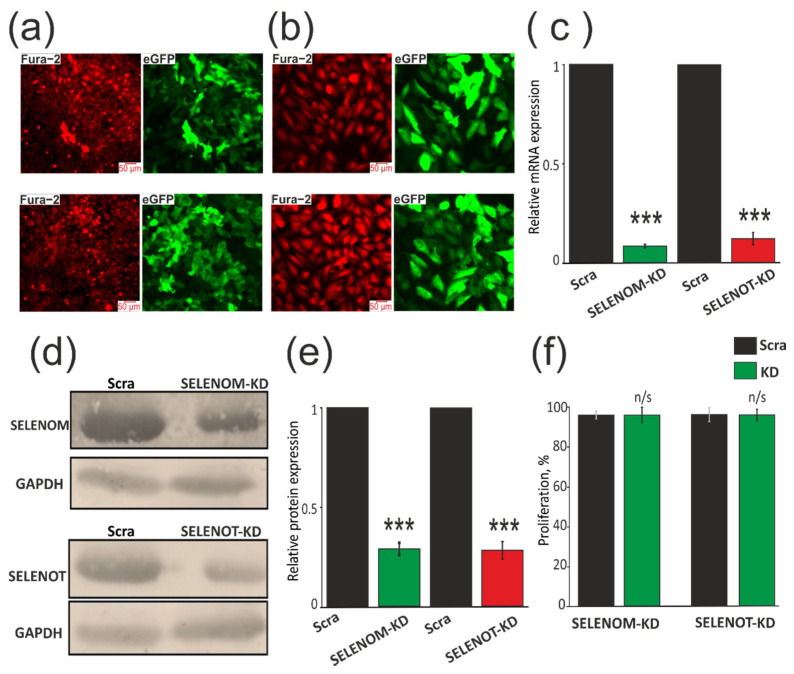
Evaluation of SELENOM-KD and SELENOT-KD efficiency by fluorescent microscopy (**a**,**b**), real-time PCR (**c**), and Western blotting (**d**,**e**). KD effect on cell proliferation. (**a**,**b**) Images of A-172 cell culture in Fura-2 fluorescence (Fura-2) at an excitation of 380 nm and eGFP fluorescence indicating efficiency of expression of the construct transduced into cells. (**c**) Real-time PCR analysis of mRNA expression of genes encoding SELENOM and SELENOT in A-172 cells transduced with constructs SELENOM-KD (or SELENOT-KD) and scrambled RNA (Scra). (**d**) Western blot analysis of SELENOM and SELENOT content in samples. (**e**) Quantification of the content of SELENOM and SELENOT in A-172 cells transduced with constructs SELENOM-KD (or SELENOT-KD) and scrambled RNA (Scra). Comparison of KD groups vs. Scra: *** *p* < 0.001. (**f**) Proliferation assay performed by MTT analysis. The optical density at 590 nm was measured, and the values of the respective untreated cells were defined as 100%. Standard deviations were determined by analysis of data from at least three independent experiments and were indicated by error bars. Comparison of KD groups vs. Scra: n/s—data not significant (*p* > 0.05).

**Figure 2 biology-11-00811-f002:**
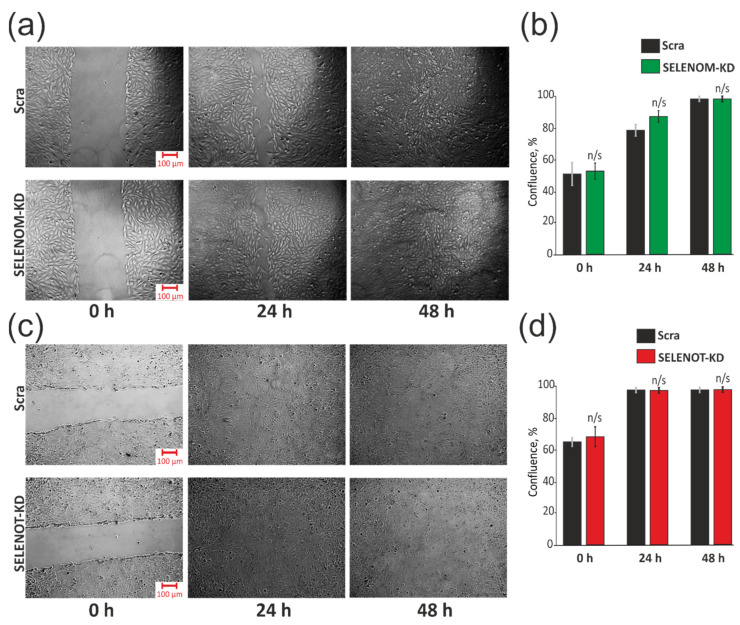
Effects of SELENOM-KD and SELENOT-KD on A-172 cells migration by the “wound healing assay”. The rate of “wound healing” and the formation of a monolayer (**a**) and quantitative assessment of the rate of “wound healing” (**b**) after 24 and 48 h in A-172 cells with SELENOM-KD, with SELENOT-KD (**c**,**d**) respectively) and in control cells (Scra). Three independent replications of this series of experiments were carried out. Comparison of KD groups vs. Scra: n/s—data not significant (*p* > 0.05).

**Figure 3 biology-11-00811-f003:**
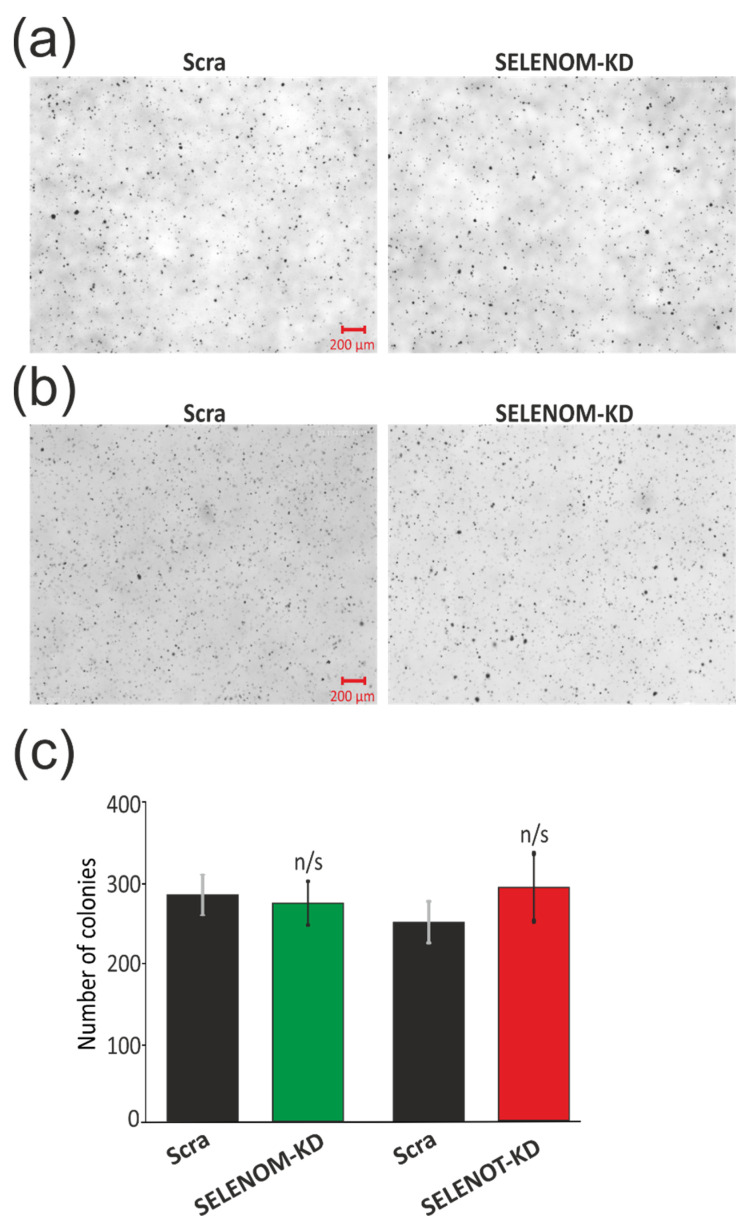
Effects of SELENOM-KD and SELENOT-KD on anchorage-independent colony formation in soft agar. Representative soft agar dishes with stained colonies for Control (Scra), SELENOM-KD (**a**), and SELENOT-KD (**b**). Quantification of grown colonies in A-172 cells with SELENOM-KD or SELENOT-KD and in control cells (Scra) (**c**). Three independent replications of this series of experiments were carried out. Comparison of KD groups vs. Scra: n/s—data not significant (*p* > 0.05).

**Figure 4 biology-11-00811-f004:**
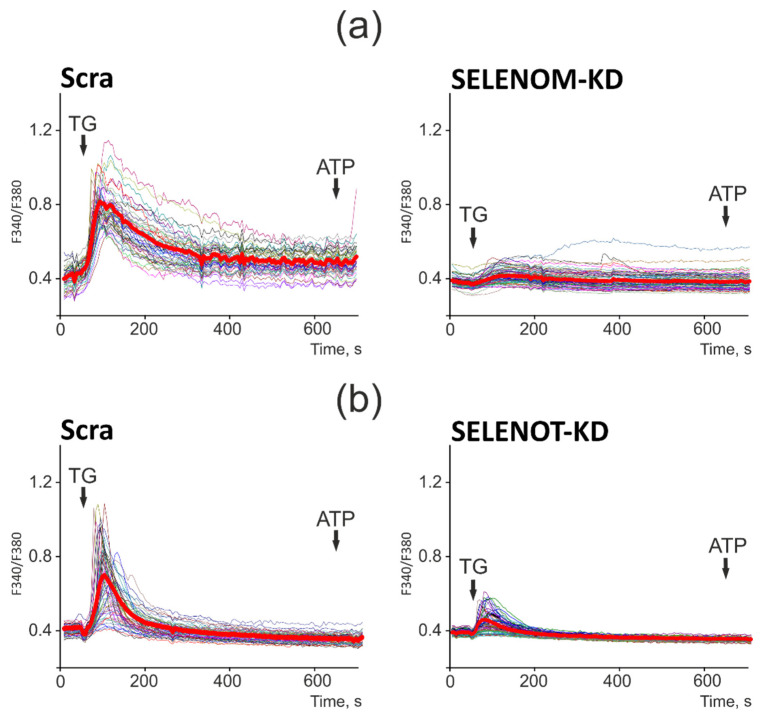
Effect of SELENOM-KD (**a**) and SELENOT-KD (**b**) on the Ca^2+^ capacity of the endoplasmic reticulum. (**a**,**b**) Ca^2+^ signals in the Scra, SELENOM-KD (**a**), and SELENOT-KD (**b**) experimental groups to the application of thapsigargin (TG, 10 mM) in a nominally calcium-free medium supplemented with Ca^2+^ chelator EGTA (0.5 mM). At the end of the experiment, 10 μM ATP was added as a test for the depletion of the ER Ca^2+^ pool. Ca^2+^ signals of cells in one experiment and their average value are presented. The number of cell passages used for experiments is 4.

**Figure 5 biology-11-00811-f005:**
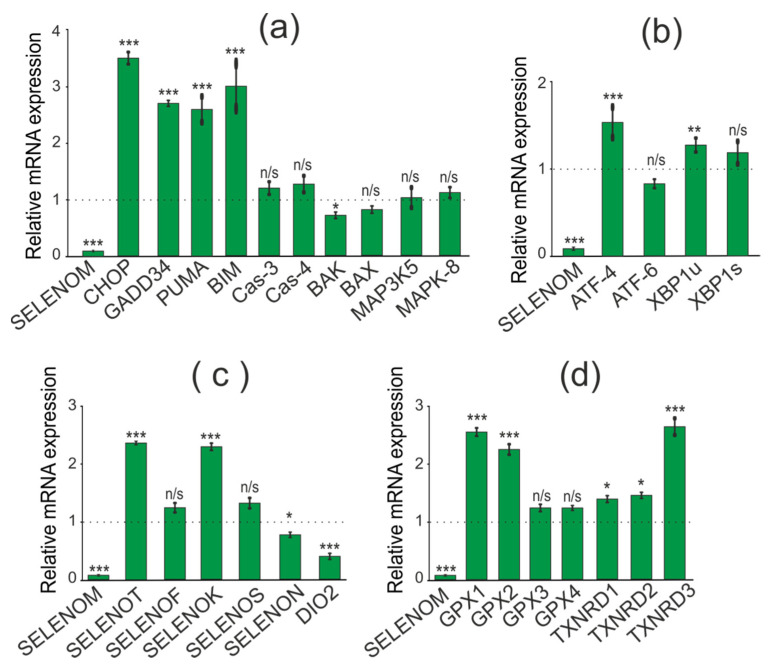
Comparative analysis of mRNA expression patterns of various genes under SELENOM-KD and without SELENOM-KD conditions, obtained using real-time PCR. (**a**) mRNA expression of pro-apoptotic genes; (**b**) mRNA expression of markers of three UPR signaling pathways; (**c**) mRNA expression of ER resident selenoproteins; (**d**) mRNA expression of selenium-containing glutathione peroxidases and thioredoxin reductases. Comparison of experimental groups regarding control, indicated as dash line (1): n/s—data not significant (*p* > 0.05), * *p* < 0.05, ** *p* < 0.01 and *** *p* < 0.001.

**Figure 6 biology-11-00811-f006:**
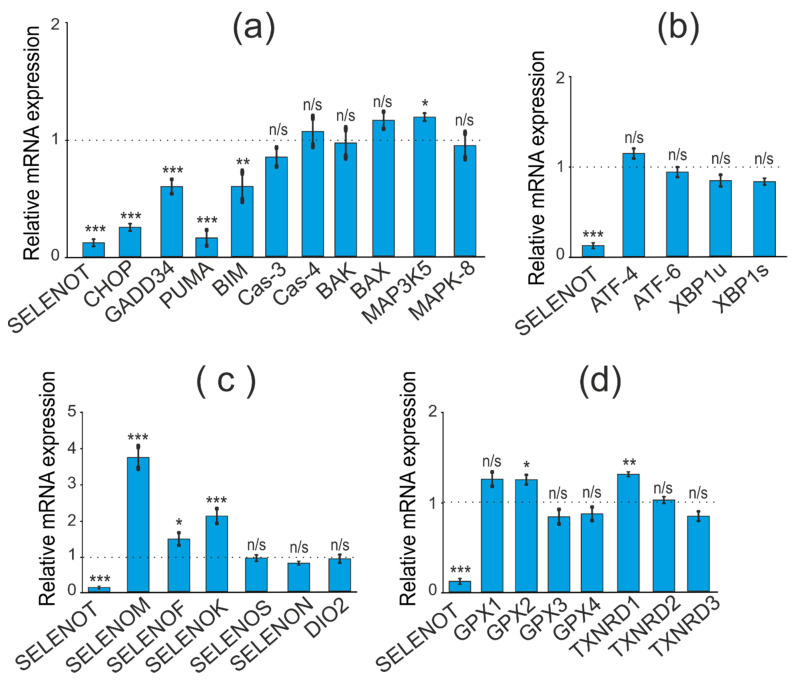
Comparative analysis of mRNA expression patterns of various genes under SELENOT-KD and without SELENOT-KD conditions, obtained using real-time PCR. (**a**) mRNA expression of pro-apoptotic genes; (**b**) mRNA expression of markers of three UPR signaling pathways; (**c**) mRNA expression of ER resident selenoproteins; (**d**) mRNA expression of selenium-containing glutathione peroxidases and thioredoxin reductases. Comparison of experimental groups regarding control, indicated as dash line (1): n/s—data not significant (*p* > 0.05), * *p* < 0.05, ** *p* < 0.01, and *** *p* < 0.001.

**Figure 7 biology-11-00811-f007:**
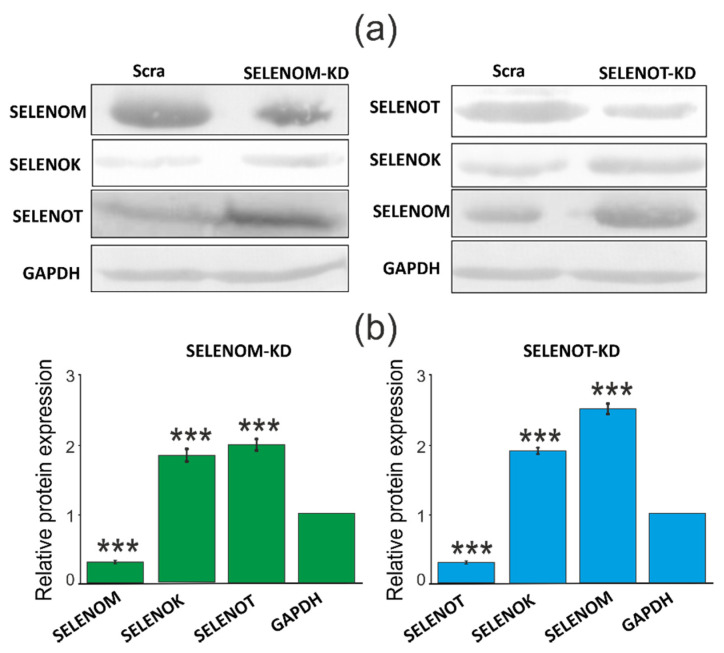
Comparative analysis of the quantitative content of ER resident selenoproteins in samples with SELENOM-KD or SELENOT-KD and control samples, performed using Western blotting (**a**). Quantification of the content of ER resident selenoproteins in A-172 cells transduced with constructs SELENOM-KD (or SELENOT-KD) and scrambled RNA (Scra) (**b**). Comparison of experimental groups regarding GAPDH level: *** *p* < 0.001.

**Figure 8 biology-11-00811-f008:**
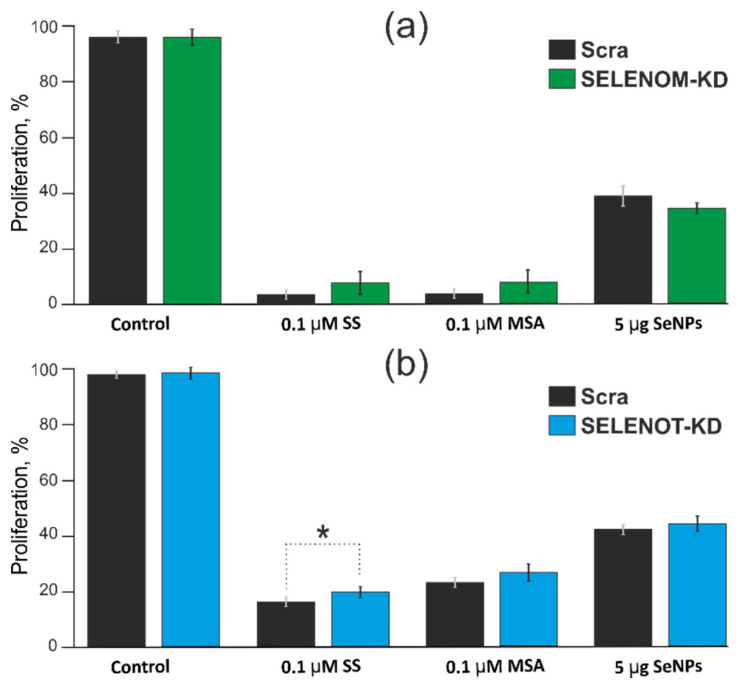
Comparison proliferation assay performed by MTT. Analysis the proliferation of A–172 cells before and after SELENOM-KD (**a**) or SELENOT-KD (**b**) after treatment with 0.1 µM SS, 0.1 µM MSA and 5 μg/mL SeNP 24 h. The optical density at 590 nm was measured, and the values of the respective untreated cells were defined as 100%. Standard deviations were determined by analysis of data from at least three independent experiments and are indicated by error bars. * *p* < 0.05.

**Figure 9 biology-11-00811-f009:**
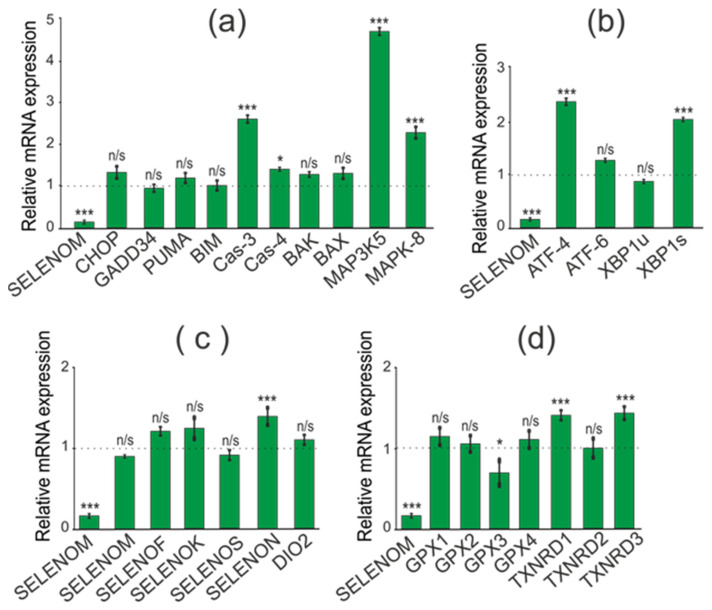
Comparative analysis of mRNA expression patterns of various genes under SELENOM-KD after 24 h treatment of A-172 cells with 5 µg/mL SeNP, obtained using real-time PCR. (**a**) mRNA expression of pro-apoptotic genes; (**b**) mRNA expression of three UPR signaling pathways markers; (**c**) mRNA expression of ER resident selenoproteins; (**d**) mRNA expression of selenium-containing glutathione peroxidases and thioredoxin reductases. Comparison of experimental groups regarding control, indicated as dash line (1): n/s—data not significant (*p* > 0.05), * *p* < 0.05 and *** *p* < 0.001.

**Figure 10 biology-11-00811-f010:**
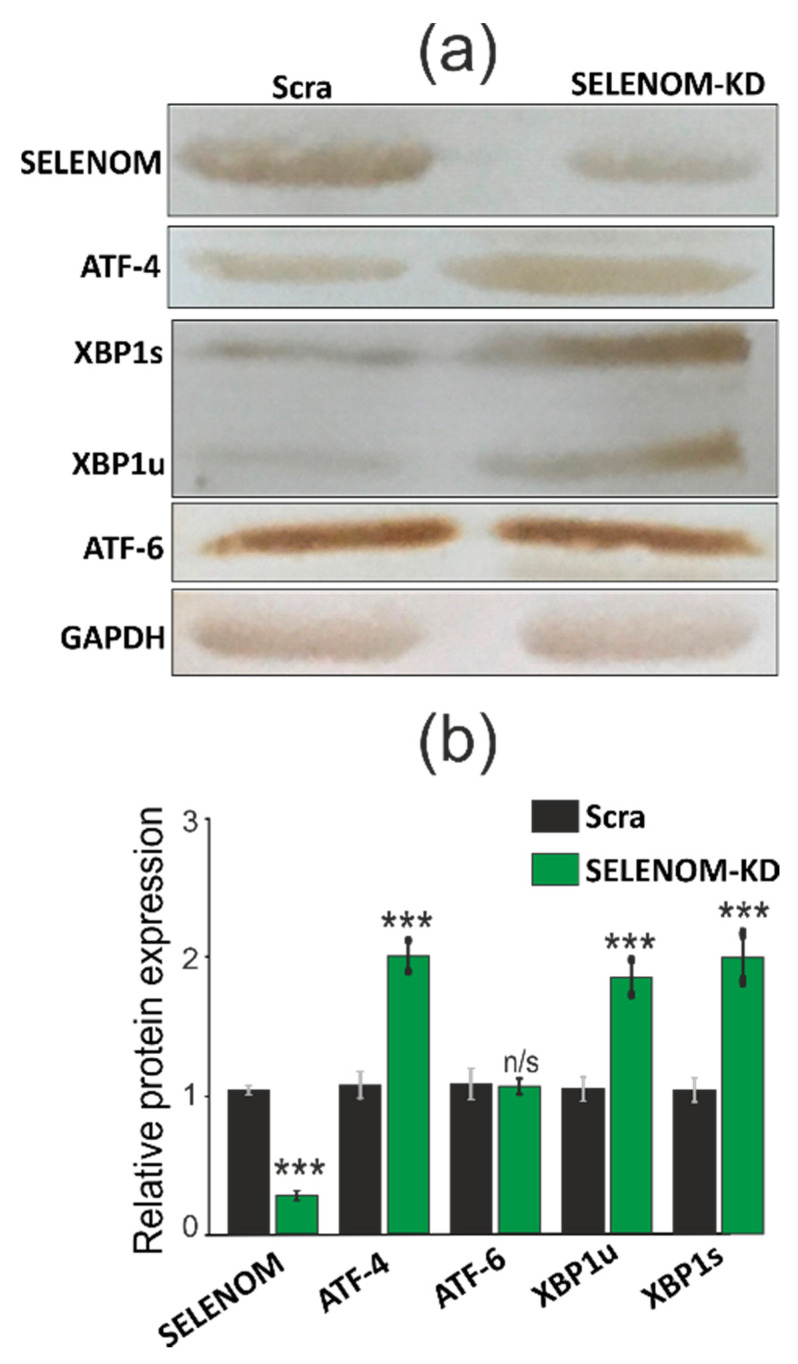
(**a**) Comparative analysis of the quantitative content UPR signaling pathways markers ATF-4, ATF-6, XBP1s, XPB1u in A-172 cells transduced with SELENOM-KD and scrambled RNA (Scra) under conditions of ER stress caused by 5 µg/mL SeNP using Western blotting. (**b**) Quantification of the content of ATF-4, ATF-6, XBP1s, XPB1u in A-172 cells transduced with SELENOM-KD and scrambled RNA (Scra) under conditions of ER stress caused by 5 µg/mL SeNP. Comparison of SELENOM-KD vs. Scra: n/s—data not significant (*p* > 0.05), *** *p* < 0.001.

**Figure 11 biology-11-00811-f011:**
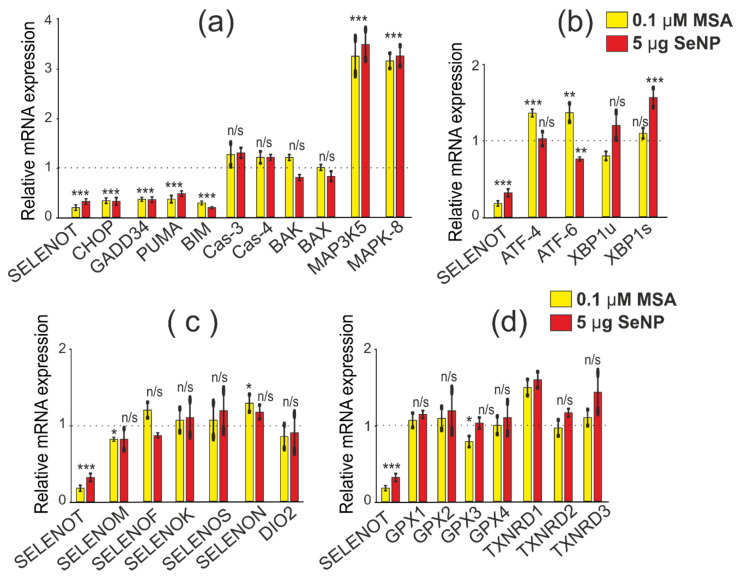
Comparative analysis of mRNA expression patterns of various genes under SELENOT-KD and after 24 h treatment of A-172 cells with 0.1 µM MSA and 5 µg/mL SeNP, obtained using real-time PCR. (**a**) mRNA expression of pro-apoptotic genes; (**b**) mRNA expression of three UPR signaling pathways markers; (**c**) mRNA expression of ER resident selenoproteins; (**d**) mRNA expression of selenium-containing glutathione peroxida44ses and thioredoxin reductases. Comparison of experimental groups regarding control, indicated as dash line (1): n/s—data not significant (*p* > 0.05), * *p* < 0.05, ** *p* < 0.01, and *** *p* < 0.001.

**Figure 12 biology-11-00811-f012:**
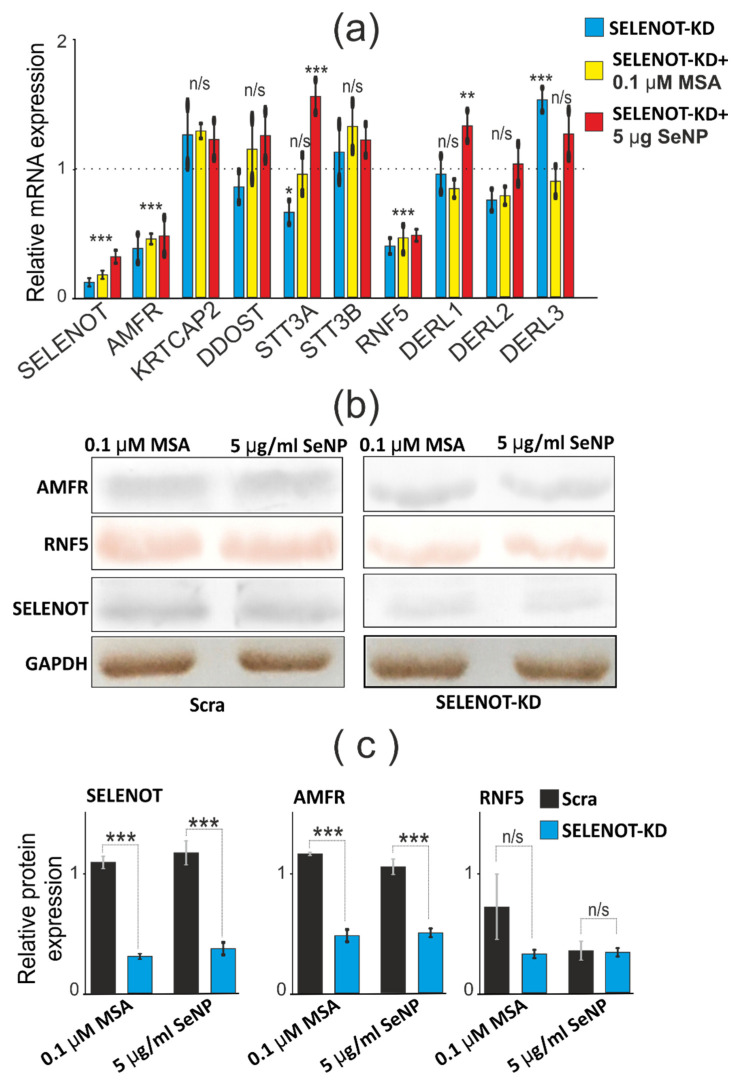
The effect of SELENOT-KD on the expression of SELENOT partner proteins under ER stress. (**a**) Comparative analysis of mRNA expression patterns of SELENOT proteins-partners under SELENOT-KD and without SELENOT-KD as well as under conditions of ER stress caused by 0.1 µM MSA or 5 µg/mL SeNP, obtained using real-time PCR. Comparison of experimental groups regarding control, indicated as dash line (1): n/s—data not significant (*p* > 0.05), * *p* < 0.05, ** *p* < 0.01 and *** *p* < 0.001; (**b**) comparative analysis of the quantitative content of SELENOT proteins-partners, using Western blotting; (**c**) quantification of the content of SELENOT proteins-partners in A-172 cells transduced with SELENOT-KD and scrambled RNA (Scra) under conditions of ER stress caused by 0.1 µM MSA or 5 µg/mL SeNP. Comparison of SELENOT-KD vs. Scra: n/s—data not significant (*p* > 0.05), *** *p* <0.001.

**Figure 13 biology-11-00811-f013:**
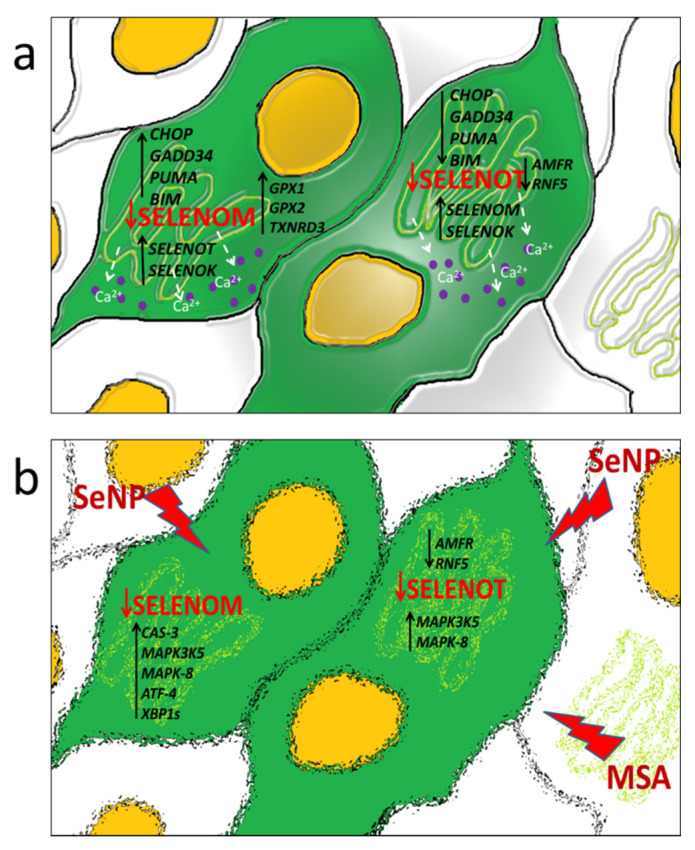
Schematic representation of the effect of SELENOM-KD and SELENOT-KD on the expression of a number of key pro-apoptotic genes, ER stress markers, functional partners of SELENOT, and calcium homeostasis in A-172 cells in the absence of exposure to ER stress inducers (**a**) and after 24 h treatment of cells with 0.1 µM MSA or 5 µg/mL SeNP (**b**).

**Table 1 biology-11-00811-t001:** Primers for real-time PCR.

Gene Name	Forward Primer 5′→3′	Reverse Primer 5′→3′
GAPDH	ACATCGCTCAGACACCATG	GCCAGTGAGCTTCCCGTT
SELENOT	TCTCCTAGTGGCGGCGTC	GTCTATATATTGGTTGAGGGAGG
SELENOM	AGCCTCCTGTTGCCTCCGC	AGGTCAGCGTGGTCCGAAG
SELENOF	TACGGTTGTTGTTGGCGAC	CAAATTGTGCTTCCTCCTGAC
SELENOK	TTTACATCTCGAACGGACAAG	CAGCCTTCCACTTCTTGATG
SELENOS	TGGGACAGCATGCAAGAAG	GCGTCCAGGTCTCCAGG
SELENON	TGATCTGCCTGCCCAATG	TCAGGAACTGCATGTAGGTGG
DIO2	AGCTTCCTCCTCGATGCC	AAAGGAGGTCAAGTGGCTG
AMFR	TCCTGGCTAGAACAAGACACC	CTGCGTAATGCCAAGAATG
KRTCAP2	GCAAAGGATTCCAAGCAAAG	TTCTTGCTCTTGCCTGTGAC
DDOST	CGCATTGATCCTTTTGTGAG	GAAGACGATGCTGAAGATGAAG
STT3A	TACAGGCAAACATATCAAGGAG	TCCAGGACATCAAGCTCAAAG
STT3B	CTACTTTGTTGAATTGCCTTATG	TTCCTTTTGGTAGTCTTCTTTG
RNF5	AAGGGCCAAATCGCGAGCG	GCGGGACAACCTTCTCTCTG
DERL1	ATTCAACTATATCATCGGAGGC	CAAGTCGAAAGCCCTGGCC
DERL2	TTCGGCCTTCTCAACTTCCAG	TGGGTCTCATTATTGGCACTG
DERL3	CTTCTCGCTGCTGCTGGGC	CCCAGGAAGCCAGGGGTC
CHOP	GCTCTGATTGACCGAATGG	TCTGGGAAAGGTGGGTAGTG
GADD34	CTCCGAGAAGGTCACTGTCC	GACGAGCGGGAAGGTGTGG
PUMA	CAGATATGCGCCCAGAGAT	CCATTCGTGGGTGGTCTTC
BIM	GGACGACCTCAACGCACAGTACGAG	GTAAGGGCAGGAGTCCCA
ATF-4	GTGTTCTCTGTGGGTCTGCC	GACCCTTTTCTTCCCCCTTG
ATF-6	AACCCTAGTGTGAGCCCTGC	GTTCAGAGCACCCTGAAGA
XBPu	ACTCAGACTACGTGCACCTC	GTCAATACCGCCAGAATCC
XBPs	CTGAGTCCGCAGCGGTGCAGG	GGTCCAAGTTGTCCAGAATG
CAS-3	GCATTGAGACAGACAGTGGTG	AATAGAGTTCTTTTGTGAGCATG
CAS-4	CACGCCTGGCTCTCATCATA	TAGCAAATGCCCTCAGCG
MAP3K5	AACACCTGAAGCTTAAGTCCC	TCAATGATAGCCTTCCACAGTG
MAPK-8	AAAGGGAACACACAATAGAAGAG	GCTGCTGCTTCTAGACTG
BAX	GGGCTGGACATTGGACTTC	AACACAGTCCAAGGCAGCTG
BAK	GAGAGTGGCATCAATTGGGG	CAGCCACCCCTCTGTGCAATCCA
GPX1	CTACTTATCGAGAATGTGGCG	CGAAGAGCATGAAGTTGGG
GPX2	CCCTTCCGACGCTACAGCCG	GGAGCCCAAGTTGAATCACC
GPX3	CCCCCACTCCTACTTCCTG	CCGAAGGAGCAGGGGTGG
GPX4	CCATGCACGAGTTTTCCG	AATTTGACGTTGTAGCCCG
TXNRD1	GGTCTGGCAGCTGCTAAGG	TAGCCCCAATTCAAAGAGC
TXNRD2	TGGGTGTGGCAGTGGGAGAC	TCCCCTGAGCCATCCCTGTG
TXNRD3	CCTTTGCTTTGTTGTTTCTGTG	TAGTGAGTGTGAGGGTGAAGC

## Data Availability

The data presented in this study are available on request from the corresponding authors.

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
