# Peer review of "The Role of Selenoproteins SELENOM and SELENOT in the Regulation of Apoptosis, ER Stress, and Calcium Homeostasis in the A-172 Human Glioblastoma Cell Line"

_biology, 2022, doi:10.3390/biology11060811_

Round 1
Reviewer 1 Report
The manuscript by Varlamova et al describes the effects of shRNA-mediated knockdown of the two selenoproteins SELENOM und SELENOT on the glioblastoma cell line A-172. They found no impact on the tested parameters of cell behavior (proliferation, wound healing, anchorage-independent growth) but a number of gene and protein expression changes and differences in calcium homeostasis.
While some of the data are interesting, the study also has clear limitations. The most critical one is the fact that all experiments were performed only in a single cell line. As always in such cases, this makes generalization problematic. Therefore, the title should read: “…… in the human glioblastoma cell line A-172” instead “glioblastoma cells” which suggest a broader scope than what the article actually shows. Alternatively, key findings should be confirmed in a second cell line.
The article is overly long and while there is no limit on the number of figures in this journal, a more condensed and focused presentation of the data would make a more pleasant read. For instance, figures 1-4 could easily be condensed to a two figures or even to single figure and parts moved to the supplement. Also for some experiments, there are separate figures for the two proteins, whereas in other figures data for both proteins are shown in different panels of the same figure.
The authors show that the expression of a number of apoptosis-regulating proteins is changed, but no evidence of actual changes in apoptosis rates is provided. If apoptosis is unchanged, then the conclusion that SELENOM has anti-apoptotic properties, while SELENOT is pro-apoptotic, is not justified.
The explanation of some methods is not clear. Figure 3 looks like a standard scratch assay that is normally used to assess cell migration. Why this requires an electron microscope and how this can determine contact inhibition is unclear. Data on transduction efficiency of the KD clones, which are shown as part of Figures 5 and 6 should already be presented in the context of figure 1.
Minor issues: Absence of significant differences (n/s)) should also be indicated in figures 2-4.
A number of proteins are mentioned without their function being at least briefly characterized.
Information provided in the introduction for SELENOM and SELENOT should be better comparable. For instance, the number of amino acids and the different protein domains are given for SELENOT but not for SELENOM, etc. The phrase in line 69 “SELENOM, the selenoprotein ER, is…” is unclear.
While the article is generally well written, there are some text editing problems, for instance, in lines 196, 226, 228, 479 to point out just a few.
Author Response
Dear Reviewer
The authors of the manuscript are grateful to you for the valuable advice and comments you made after a thorough analysis of our work. We have tried to fix the errors and shortcomings you mentioned in the work.
While some of the data are interesting, the study also has clear limitations. The most critical one is the fact that all experiments were performed only in a single cell line. As always in such cases, this makes generalization problematic. Therefore, the title should read: “…… in the human glioblastoma cell line A-172” instead “glioblastoma cells” which suggest a broader scope than what the article actually shows. Alternatively, key findings should be confirmed in a second cell line.
Reply. A change was made to the title of the article and clarifications throughout the text about the fact that the work was performed on the example of one A-172 cancer cell line.
The article is overly long and while there is no limit on the number of figures in this journal, a more condensed and focused presentation of the data would make a more pleasant read. For instance, figures 1-4 could easily be condensed to a two figures or even to single figure and parts moved to the supplement. Also for some experiments, there are separate figures for the two proteins, whereas in other figures data for both proteins are shown in different panels of the same figure.
Reply. Thank you for your comments that helped improve our work. The number of figures has been reduced. Micrographs added to supplementary.
The authors show that the expression of a number of apoptosis-regulating proteins is changed, but no evidence of actual changes in apoptosis rates is provided. If apoptosis is unchanged, then the conclusion that SELENOM has anti-apoptotic properties, while SELENOT is pro-apoptotic, is not justified.
Reply. We fully agree with you that additional experiments are required to study the rate of apoptosis over time under conditions of reduced activity of these selenoproteins. Therefore, in the «Conclusions» section, appropriate explanations were made.
The explanation of some methods is not clear. Figure 3 looks like a standard scratch assay that is normally used to assess cell migration. Why this requires an electron microscope and how this can determine contact inhibition is unclear. Data on transduction efficiency of the KD clones, which are shown as part of Figures 5 and 6 should already be presented in the context of figure 1.
Reply. Thanks for the comments. We did not use an electron microscope in our studies. This is a text error. Corrected. We used a Zeiss Axiovert 200 M light microscope. The figures have been corrected according to your comment.
Minor issues: Absence of significant differences (n/s)) should also be indicated in figures 2-4.
Reply. Corrected
A number of proteins are mentioned without their function being at least briefly characterized.
Reply. In the Discussion section, additional information was added about the functions of some proteins (CHOP, BIM, PUMA, GPX1, GPX2 and others).
Information provided in the introduction for SELENOM and SELENOT should be better comparable. For instance, the number of amino acids and the different protein domains are given for SELENOT but not for SELENOM, etc. The phrase in line 69 “SELENOM, the selenoprotein ER, is…” is unclear.
Reply. The information about the investigated selenoproteins M and T was brought into line, reducing part of the material about T and supplementing the information about M by analogy with T. In line 69, a change was made, simplifying the proposal.
Reviewer 2 Report
Overall, this paper demonstrated that the function of selenoproteins SELENOM and SELENOT in human glioblastoma cells, including the mechanisms of regulation of the processes of apoptosis, cell proliferation, and ER-stress, which is important for future clinical studies. The experiments were well designed and executed. It would be beneficial for the audience who is interested in brain pathology studies.
Minor points:
- The authors missed scale bars in some figures: Figure 5, Figure 6.
Author Response
Dear Reviewer,
The authors of the manuscript are grateful to you for the valuable advice and comments you made after a thorough analysis of our work. We have tried to fix the errors and shortcomings you mentioned in the work.
While the article is generally well written, there are some text editing problems, for instance, in lines 196, 226, 228, 479 to point out just a few.
Reply. We have corrected grammatical and spelling errors throughout the text of the manuscript.
Minor points:
- The authors missed scale bars in some figures: Figure 5, Figure 6.
Reply. Corrected
Reviewer 3 Report
1) Although the standard of English is reasonably good, the majority of sentences are excessively long and convoluted. Communication of ideas will be vastly improved if the manuscript is re-written, using shorter and more efficient sentences. For example, the single sentence paragraph between lines 64 and 68 could be rephrased as two sentences: "Among them, the least studied are the two selenoproteins SELENOM and SELENOT, which belong to a thioredoxin fold-containing family of transmembrane proteins. These differ from classical thioredoxins in the arrangement of secondary structural elements, the presence of additional α-helices, and of a selenocysteine in the active center." It would be good if the spelling and grammar were checked throughout, eg. to a native English speaker, the phrase "double cross" (line 86) means to betray, rather than to cross twice. In Section 2.6, the language switches from past to present tense, then back again to past tense.
2) "Section 2.5. MTT analysis." The MTT assay does not directly measure cell viability or proliferation. It is better to describe this as a measure of mitochondrial reducing power, which does depend on cell number and viability, but also other factors, eg. mitochondrial density per cell.
3) "Section 2.7. Analysis of the ability of cancer cells....". This assay is widely referred to as the "wound healing assay". It is not a measure of contact inhibition, but of collective cell migration (this might be influenced by contact inhibition, but it is certainly not the same thing).
4) "Section 2.8. Registration of changes in intracellular Ca2+". The description of this method would not allow a reader to repeat this experiments. For example, what concentration of fura-2-acetoxynethyl ester (described as Fura-2, which is a free acid and could not diffuse into cells) was used and for how long and at what temperature was it loaded? What detection system was used, eg. a CCD camera or a photomultiplier? How were images acquired and at what acquisition rate? It seems like the processing of the fura-2 data is described in Section 2.9 (although this was not immediately clear), but this description is inadequate and contains errors. For example, fura-2 data can be expressed as either a ratio of light intensity emitted at an excitation wavelength of 340 nm divided by that excited at 380 nm (as it is in this manuscript), or can be calibrated to absolute [Ca2+], but not "probe fluorescent units". Was any background subtraction performed? Also, it is better to describe the measurements using fura-2 as "Cytoplasmic Ca2+" rather than "Intracellular Ca2+" as the dye should be confined to the cytoplasmic (as it appears to be in Fig. 5a) and 6a).
5) Section 3.1, line 312. Did the authors really use an electron microscope to acquire images in the wound healing assay?
6) Figures 3, 4, 5a and 6a. The quality of the brightfield micrographs shown in these images is poor. The contrast is so low, that it is difficult to observe features in these images.
7) Why was ATP added prior to the end of the fura-2 experiments? In most cells, it has little effect on cytoplasmic Ca2+. Better positive control experiments would either be add-back of Ca2+ (1 or 2 mM) to the extracellular medium, or addition of a Ca2+ ionophore.
8) Section 3.2, lines 350 to 352. There is an unsubstantiated claim that the decline in cytoplasmic Ca2+ following the peak resulting from addition of thapsigargin is due to the activity of plasmalemma calcium ATPases (PMCA). It is most likely that this decline in cytoplasmic Ca2+ is due to the concerted actions of multiple components: cytoplasmic Ca2+ buffers, sodium-calcium exchangers, mitochondrial Ca2+ uptake and the action of PMCAs. However, none of this can be stated definitively for this system, without experimental evidence.
9) The phrase "but in non-excitable cells, such as A-172 cells, the key event in calcium signaling is the mobilization of Ca2+ ions from the ER", is misleading. Ca2+ influx plays an important part in Ca2+ signalling in non-excitable cells. This can occur through a wide variety of cation channels, include transient receptor potential (TRP) channels, store-operated calcium entry channels (eg. Orai1-3) and ATP receptors (P2X).
Author Response
Dear Reviewer 3,
The authors of the manuscript are grateful to you for the valuable advice and comments you made after a thorough analysis of our work. We have tried to fix the errors and shortcomings you mentioned in the work.
1) Although the standard of English is reasonably good, the majority of sentences are excessively long and convoluted. Communication of ideas will be vastly improved if the manuscript is re-written, using shorter and more efficient sentences. For example, the single sentence paragraph between lines 64 and 68 could be rephrased as two sentences: "Among them, the least studied are the two selenoproteins SELENOM and SELENOT, which belong to a thioredoxin fold-containing family of transmembrane proteins. These differ from classical thioredoxins in the arrangement of secondary structural elements, the presence of additional α-helices, and of a selenocysteine in the active center." It would be good if the spelling and grammar were checked throughout, eg. to a native English speaker, the phrase "double cross" (line 86) means to betray, rather than to cross twice. In Section 2.6, the language switches from past to present tense, then back again to past tense.
Reply. The text has been edited, all translation errors have been corrected.Mistranslation in line 86 corrected, in section 2.6. time has been adjusted.
2) "Section 2.5. MTT analysis." The MTT assay does not directly measure cell viability or proliferation. It is better to describe this as a measure of mitochondrial reducing power, which does depend on cell number and viability, but also other factors, eg. mitochondrial density per cell.
Reply. We agree with your comment that MTT is a colorimetric method for measuring mitochondrial reductase activity in living cells, so the incorrect information in Section 2.5 has been removed.
3) "Section 2.7. Analysis of the ability of cancer cells....". This assay is widely referred to as the "wound healing assay". It is not a measure of contact inhibition, but of collective cell migration (this might be influenced by contact inhibition, but it is certainly not the same thing).
Reply. We fully agree with your comment regarding the incorrect name and interpretation of the method in section 2.7, so we have corrected the name to the generally accepted "wound healing assay". In addition, we made changes in the results and discussion section, replacing the phrase "degree of contact inhibition" with "degree of collective cell migration"
4) "Section 2.8. Registration of changes in intracellular Ca2+". The description of this method would not allow a reader to repeat this experiments. For example, what concentration of fura-2-acetoxynethyl ester (described as Fura-2, which is a free acid and could not diffuse into cells) was used and for how long and at what temperature was it loaded? What detection system was used, eg. a CCD camera or a photomultiplier? How were images acquired and at what acquisition rate? It seems like the processing of the fura-2 data is described in Section 2.9 (although this was not immediately clear), but this description is inadequate and contains errors. For example, fura-2 data can be expressed as either a ratio of light intensity emitted at an excitation wavelength of 340 nm divided by that excited at 380 nm (as it is in this manuscript), or can be calibrated to absolute [Ca2+], but not "probe fluorescent units". Was any background subtraction performed? Also, it is better to describe the measurements using fura-2 as "Cytoplasmic Ca2+" rather than "Intracellular Ca2+" as the dye should be confined to the cytoplasmic (as it appears to be in Fig. 5a) and 6a).
Reply. Thanks for the very valuable comments. The necessary information is added to the material and methods.
5) Section 3.1, line 312. Did the authors really use an electron microscope to acquire images in the wound healing assay?
Reply. An optical rather than an electron microscope was used for the "wound healing assay", the correction was made in section 2.7.
6) Figures 3, 4, 5a and 6a. The quality of the brightfield micrographs shown in these images is poor. The contrast is so low, that it is difficult to observe features in these images.
Reply. Image contrast has been improved. The size of the micrographs has been enlarged. The figures have been changed and partially added to the supplementary.
7) Why was ATP added prior to the end of the fura-2 experiments? In most cells, it has little effect on cytoplasmic Ca2+. Better positive control experiments would either be add-back of Ca2+ (1 or 2 mM) to the extracellular medium, or addition of a Ca2+ ionophore.
Reply. At the end of the experiment, ATP was added at a concentration of 10 μM as a test for the depletion of the Ca2+ pool of the endoplasmic reticulum. In principle, such an application of ATP was not required, since thapsigargin is a selective SERCA blocker. Since there was no Ca2+ in the extracellular medium, we would also not see a signal in response to ionomycin. But we agree about the positive control in the form of application of 2 mM Ca2+. This application would have shown activation of Ca2+ entry from the outside, which, however, was not the aim of the work. Changes have been made to the figure legend.
8) Section 3.2, lines 350 to 352. There is an unsubstantiated claim that the decline in cytoplasmic Ca2+ following the peak resulting from addition of thapsigargin is due to the activity of plasmalemma calcium ATPases (PMCA). It is most likely that this decline in cytoplasmic Ca2+ is due to the concerted actions of multiple components: cytoplasmic Ca2+ buffers, sodium-calcium exchangers, mitochondrial Ca2+ uptake and the action of PMCAs. However, none of this can be stated definitively for this system, without experimental evidence.
Reply. We express our gratitude to the expert for the comment. Agree. Indeed, it is. Information about the role of PMCA has been removed, since no additional experiments were performed to identify the mechanism for the removal of Ca2+ ions from the cytoplasm.
9) The phrase "but in non-excitable cells, such as A-172 cells, the key event in calcium signaling is the mobilization of Ca2+ ions from the ER", is misleading. Ca2+ influx plays an important part in Ca2+ signalling in non-excitable cells. This can occur through a wide variety of cation channels, include transient receptor potential (TRP) channels, store-operated calcium entry channels (eg. Orai1-3) and ATP receptors (P2X).
Reply. Thank you. Agree with the reviewer. Since this work was not intended to investigate the role of selenoprotein knockdowns on the mechanisms of the increase in Ca2+ ions in the cell cytoplasm, we rewrote this sentence more carefully.
Round 2
Reviewer 1 Report
The authors have tried to address all concerns. In part, however, this has created (minor) new problems, that need to be fixed.
The title now better reflects the content of the manuscript, but the English grammar is not correct. It needs to be either “in the A-172 human glioblastoma cell line” or “in human A-172 human glioblastoma cells”.
Generally, some English language editing is still required – especially in the newly added parts.
Figure 15 should now be Figure 12.
Author Response
Dear Reviewer,thank you for your comments.
- We have corrected the title of the manuscript, replacing it with the following: «The role of selenoproteins SELENOM and SELENOT in the regulation of apoptosis, ER-stress, and calcium homeostasis in the A-172 human glioblastoma cell line».
- The text of the manuscript was checked for grammatical and spelling errors, and corrections were made.
- The numbering of the figures has been corrected to reflect the added figures, which was done on the recommendation of another Reviewer.
Reviewer 3 Report
Although the manuscript is considerably improved, removal of the micrographs showing colony formation and wound healing does not constitute correction of their low quality. Quantitative data based on brightfield micrographs should include representative examples of these micrographs.
Author Response
Dear Reviewer, thank you for your comment.
The authors returned to the text of the manuscript improved photographs reflecting the results of two tests.